# Evaluation of 73 Enlisted Patients for Liver Transplant with Unknown Etiology Reveals a Late-Diagnosed Case of Lysosomal Acid Lipase Deficiency

**DOI:** 10.3390/ijms25168648

**Published:** 2024-08-08

**Authors:** Karina Lucio de Medeiros Bastos, Bruno de Oliveira Stephan, Bianca Domit Werner Linnenkamp, Larissa Athayde Costa, Fabiana Roberto Lima, Laura Machado Lara Carvalho, Rachel Sayuri Honjo, Uenis Tannuri, Ana Cristina Aoun Tannuri, Chong Ae Kim

**Affiliations:** 1Unidade de Genética, Instituto da Criança, Hospital das Clínicas, Faculdade de Medicina da Universidade de São Paulo (FMUSP), Sao Paulo 05403-000, SP, Brazillarissasathayde@gmail.com (L.A.C.);; 2Departamento de Anatomia Patológica, Hospital das Clínicas, Faculdade de Medicina da Universidade de São Paulo (FMUSP), Sao Paulo 05403-010, SP, Brazil; 3Centro de Estudos do Genoma Humano e Células-Tronco, Instituto de Biociências da Universidade de São Paulo (IBUSP), Sao Paulo 05508-090, SP, Brazil; 4Cirurgia Pediátrica, Instituto da Criança, Hospital das Clínicas, Faculdade de Medicina da Universidade de São Paulo (FMUSP), Sao Paulo 05403-000, SP, Brazil

**Keywords:** lysosomal acid lipase deficiency, *LIPA* gene testing, liver transplant, genetic counseling

## Abstract

Lysosomal acid lipase deficiency (LALD) varies from a severe infantile-onset form (Wolman disease) to a late-onset form known as cholesteryl ester storage disease (CESD), both of which are autosomal recessive disorders caused by biallelic *LIPA* pathogenic variants. We evaluated seventy-three patients enlisted for liver transplant (LT) at Instituto da Criança (HCFMUSP—Brazil) who were subjected to LAL activity measurement and *LIPA* Sanger sequencing analysis, resulting in a positive LALD diagnosis for only one of these individuals. This LALD patient presented recurrent diarrhea, failure to thrive, hepatomegaly, and dyslipidemia at the age of 4 months and liver failure by the age of 13 years. The LALD diagnosis confirmation was conducted at 24 years old due to low levels of LAL enzyme activity. The causal homozygous variant *LIPA*(NM_000235.4):c.266T>C(p.Leu89Pro) was identified, but the patient had already undergone his first LT at 18 years with several rejection episodes. Despite beginning treatment with sebelipase alfa at 26 years old (total of five infusions), this patient died at 28 years from complications after his second liver transplant. LALD is an important differential diagnosis in cases presenting with hepatomegaly, elevated liver enzymes, and dyslipidemia. Detecting low/absent LAL activity and identifying the *LIPA* causal variant are essential for diagnosis and specific treatment, as well as for appropriate genetic counseling. Early diagnosis, along with sebelipase alfa therapy, may improve the prognosis of affected patients.

## 1. Introduction

The clinical spectrum of lysosomal acid lipase deficiency (LALD) ranges from the severe early-onset (infantile) form, known as Wolman disease (WD—OMIM#620151), characterized by ≤1% of lysosomal acid lipase (LAL) activity, to the late-onset form (childhood/adult-onset), referred to as cholesteryl ester storage disease (CESD—OMIM#278000), in which patients retain up to 10% enzymatic activity. Both conditions are autosomal recessive disorders caused by biallelic pathogenic variants in the *LIPA* (lysosomal acid lipase A) gene. LALD is marked by ectopic lysosomal lipid accumulation [1]. The diminished LAL activity leads to a gradual accumulation of cholesteryl esters and triglycerides in hepatocytes, adrenal glands, intestines, and macrophage–monocyte system cells [2], leading to premature organ damage and high precocious mortality [1].

LALD is an extremely rare condition, with an estimated global prevalence of 1 in 177,000 based on a meta-analysis [3]. Meanwhile, in a study regarding the frequency of lysosomal storage diseases (LSD) in Brazil, the prevalence of LALD was estimated at 0.011 per 100,000 births [4]. However, in the scientific literature, there are only a few Brazilian cases described [5,6,7]. It is an underdiagnosed disease, probably with a high number of unreported cases misdiagnosed as uncharacteristic fatty liver disease [2,8,9].

Here, we describe an adult patient diagnosed with LALD at 24 years old. Despite showing initial clinical signs since 4 months of age, he was misdiagnosed with Niemann–Pick type B (OMIM# 607608; acid sphingomyelinase deficiency) based on enzymatic testing and pathological findings. At 18, he underwent his first liver transplant (LT). Unfortunately, after experiencing multiple rejection episodes, he died at the age of 28 after a second LT.

## 2. Cohort Study That Led to the Diagnosis of the Reported Patient

The workflow presented in Figure 1 summarizes our strategy for investigating LALD in a cohort of patients undergoing LT with unknown etiology. We evaluated 54 out of 224 patients who underwent LT due to hepatic failure of unknown etiology from January 2015 to July 2019 at Instituto da Criança (HCFMUSP, Brazil), a recognized reference center for LT in Latin America [10]. Additionally, we included another 19 patients from the waiting list.

The 73 selected patients had a diagnosis of cirrhosis or were transplanted due to cirrhosis without a defined etiology (cryptogenic). The diagnosis of cirrhosis was based on the presence of grade-4 fibrosis on liver biopsy or signs of cirrhotic portal hypertension identified by laboratory and imaging exams. For the diagnosis of cryptogenic cirrhosis, patients underwent extensive investigation, including viral serologies, alpha-1-antitrypsin, ceruloplasmin, urinary copper, immunoglobulins, autoantibodies for autoimmune liver diseases, protein electrophoresis, and imaging exams.

A LALD investigation was conducted for the 73 selected patients. Blood samples from these patients were used to perform a LAL enzyme activity assay using the fluorometric substrate 4-methylumbelliferyl palmitate [11], primarily conducted at the Children’s Department of Laboratories in Seattle (USA), but for some cases at Hospital das Clínicas in Porto Alegre (Brazil). The only patient with low LAL activity underwent *LIPA* gene analysis via Sanger sequencing to identify the causal variant.

All research participants or their legal guardians signed a consent form. This study is part of a project approved by the institutional ethics committee, and the procedures comply with the Helsinki Declaration.

## 3. Case Report

The positive LALD patient in our cohort study is the second child of a consanguineous couple. He was born by cesarean section, weighing 3400 g, after an uneventful pregnancy.

He had diarrhea since four months of age, which escalated to a chronic state, ultimately resulting in failure to thrive by the eighth month. Despite discontinuing breastfeeding and introducing soy milk along with various supplements, the child continued to present inadequate weight gain. During that time, even though his neurodevelopment was mostly adequate for his age, laboratory tests revealed hypercholesterolemia (total cholesterol = 428 mg/dL; LDL = 374 mg/dL; HDL = 16 mg/dL; triglycerides = 190 mg/dL), and hepatomegaly became evident. More comprehensive testing confirmed the elevation of several hepatic enzymes, such as aspartate transaminase (AST) = 87 U/L (3× the upper limit of normal—ULN); alanine transaminase (ALT) = 144 U/L (4× ULN); gamma-glutamyltransferase (GGT) = 59 U/L (2× ULN); and alkaline phosphatase (ALP) = 1014 U/L (4× ULN). Fortunately, other hepatic function tests (such as albumin and coagulation tests) remained mostly within reference values, as well as glycemic levels and complete blood count (CBC) results.

Given the suspicion of an inborn error of metabolism, a liver biopsy was performed, revealing enlarged portal tracts with mild fibrosis and septa, microvesicular steatosis, and macrophages with cytoplasmic microvacuolization; all these findings were compatible with the hypothesis of Niemann–Pick disease. Additionally, the acid sphingomyelinase activity determination in leukocytes was low (53.9 nmol/h/mg, below the normal range of 86 to 173), and acid phosphatase was high (37.1 U/L, above the normal range of 0.5 to 11), reinforcing the possibility of type-B Niemann–Pick disease. However, electron microscopy showed negative images for cholesterol crystals in hepatocytes and histiocytes.

With the worsening of chronic diarrhea and failure to thrive, the child’s condition deteriorated, along with several hospitalizations due to asthma and bronchospasms. Eventually, splenomegaly and earlier signs of cirrhosis took place; endoscopy showed esophageal varicosities, duodenal polyps, and a confluence of mucous anomalies suggestive of deposits of cholesterol. A duodenal biopsy confirmed thickening of the intestinal villi due to numerous xanthomatous histiocytes in the lamina propria and even in the submucosa (to a lesser degree). Overall, functional coprology was normal.

A thoracic computed tomography scan revealed interstitial pneumopathy in the airway, also suggesting a deposit disease; spirometry indicated a restrictive disorder; and the echocardiogram with microbubbles was positive for the presence of an intrapulmonary shunt. Finally, arterial blood gas analysis confirmed hypoxemia (oxygen pressure: 68.7 mmHg; saturation: 93.6%), strongly suggesting hepatopulmonary syndrome.

As the hepatopathy and portal hypertension progressed, hospitalizations due to ascites became recurrent, with worsening breathing disability; at this point, by the age of 14, preparations for an LT officially began (MELD = 17). While waiting for a cadaveric donor, the patient developed hypoxemia and was eventually relisted with priority due to the advanced stage of hepatopulmonary syndrome.

Finally, at 18 years old, he received an LT from a deceased donor. Histopathological analysis of his explanted liver showed micronodular cirrhosis, some needle-shaped clefts inside foamy Kupffer cells, and typical signs of LALD, considering the finding of cholesteryl crystals (Figure 2).

After this LT, the patient immediately developed acute cellular rejection, which was readily controlled with specific therapy and left no damage to the graft. Under the care of the Gastroenterology Unit, he was then transferred to the adult hepatic transplant team. His cholesterol also remained stable overall, with an elevation of HDL. He still evolved with a reduction in splenomegaly of nearly 50% and remarkable improvement to his respiratory condition.

However, two years after the initial transplant, the patient experienced acute cellular rejection, which quickly turned into early chronic rejection. Within the next three years, a sudden rise in transaminase levels prompted a new liver biopsy, which revealed chronic ductopenia and signs of fibrosis, even though there was no evidence of recurrence of the original disease.

After four years, a duodenal biopsy showed multiple xanthomatous macrophages in the lamina propria. Since hepatic enzymes remained elevated, a new liver biopsy was performed the following year, confirming the presence of xanthomatous macrophages in the portal tracts and fibrosis with bridging, along with chronic rejection and rare sinusoidal foam cells, strongly suggesting recurrence of the storage disorder (Figure 3).

Sanger sequencing revealed a likely pathogenic homozygous missense variant in the patient’s *LIPA* gene, namely, NM_000235.4:c.266T>C (p.Leu89Pro) (Appendix A), classified according to the American College of Medical Genetics and Genomics (ACMG) guidelines [12,13], considering the criteria and arguments presented in Appendix A. Thus, the patient’s diagnosis of LALD was confirmed at 24 years old based on low enzyme activity (10 pmol/hr/spt; reference interval 400 to 600) in blood spots and the detection of the *LIPA* causal variant.

The treatment with sebelipase alfa [14]—a medication containing recombinant human LAL produced in transgenic chicken eggs [15]—began at 26 years. This treatment was applied through compassionate use in partnership with Alexion Pharmaceuticals, with a total of five infusions at a dose of 1 mg/kg body weight every other week, but the patient continued to exhibit abnormal hepatic enzyme levels and, despite undergoing immunosuppressive management, progressed to develop graft fibrosis.

A year later, the liver biopsy showed chronic ductopenic rejection, advanced fibrosis associated with late acute rejection, and the presence of aggregated macrophages and xanthomatous Kupffer cells, again suggesting recurrence of the storage disorder.

Given the progressive decline of his hepatic function, the patient was enlisted for a retransplant at the age of 26. While waiting, he developed encephalopathy secondary to infection. He underwent a second liver transplant that same month, but unfortunately, he died two years later at the age of 28, following a two-month hospitalization due to encephalopathy and complications arising from the transplant.

## 4. Discussion

LAL is an enzyme expressed by all cell types except erythrocytes, responsible for hydrolyzing fatty acid–glycerol esters and cholesteryl esters in the lysosome at acidic pH. Pathogenic variants in the gene that encodes this enzyme (*LIPA*) lead to ectopic lysosomal lipid accumulation primarily in the liver, intestine, spleen, adrenal glands, lymph nodes, bone marrow, and macrophages, causing LALD [1], which has significant morbidity and mortality, primarily due to liver-related complications and cardiovascular disease [2].

Because LALD is a life-threatening disease, early diagnosis is essential for management and survival, but the diagnosis is particularly difficult due to the lack of pathognomonic clinical signs and shared similarities with other pathologies [1]. LALD should be considered when patients present with dyslipidemia, elevated aminotransferase levels, and liver abnormalities of unknown etiology [1,16].

The identification of low LAL activity and/or the presence of biallelic deleterious variants in the *LIPA* gene is essential for diagnostic confirmation [17]. Although molecular confirmation through *LIPA* gene sequencing is always recommended, LAL activity measurement in leukocytes, fibroblasts, or dried blood spots (DBS) can also be a cost-effective alternative way to confirm the diagnosis [1].

Since it is an autosomal recessive disease, the occurrence of parental consanguinity should also be considered as supporting the hypothesis of LALD. However, the absence of parental consanguinity should not exclude the LALD hypothesis. Only 19% of cases result from parental consanguinity, according to the systematic review conducted by Witeck et al. (2022) [2]. In cases of a positive diagnosis for LALD, there is a 25% risk of recurrence for future offspring of the patient’s parents, and this is important information for genetic counseling.

The low frequency of LALD [3,4] was corroborated in our cohort study (just one positive case among 73 patients undergoing liver transplant with unknown etiology). This rarity is also a contributing factor to the diagnostic challenge, as clinicians often have not had the opportunity to gain the experience necessary to understand the specificities of this disease. The case report we are presenting also illustrates the relevance of accurate and early diagnosis of LALD, as our patient was initially diagnosed with Niemann–Pick type B, received the correct diagnosis only at 24 years old, and died at 28 after the second liver transplant.

Interestingly, our patient’s mutation—NM_000235.4:c.266T>C(p.Leu89Pro)—was absent in large population databases, both global (gnomAD) [14] and local (ABraOM) [15], but was previously submitted to ClinVar [18] (ID: 1685367), and although the submission was by a Brazilian private genetic diagnostic company, it was found that the patient was part of a neonatal screening Croatian project.

It is important to mention that the “patient 4” briefly reported by Benevides et al. is the same patient we are reporting in this manuscript. However, we now provide a detailed description of the cohort study and the diagnostic process, along with additional clinical details, including the second transplant and the therapeutic attempt with sebelipase alfa. We also emphasize the importance of early diagnosis for therapeutic success, particularly in this case, which had a fatal outcome. This additional information expands upon the previous report, offering a more comprehensive understanding of the patient’s disease phenotype and clinical course.

LALD has two clinical manifestations: WD and CESD. WD is typically characterized by poor absorption in the first days of life, resulting in malnutrition and the accumulation of cholesterol esters and triglycerides in hepatic macrophages; with time, hepatomegaly (with or without splenomegaly) and adrenocortical calcifications can lead to both hepatic and adrenal insufficiency [17] Overall, life expectancy without proper treatment does not exceed 1 year [17,19].

CESD signs frequently start later, but sometimes it can manifest early, similar to WD. Dyslipidemia (with elevated LDL and triglyceride levels and reduced HDL levels), hepatosplenomegaly, and elevated transaminases are frequent, alongside failure to thrive, anemia, atherosclerosis, cirrhosis, portal hypertension, and other chronic complications [2,9,17]. Although life expectancy is variable, there is an increased risk of hepatocarcinoma (particularly in cirrhotic patients) [17].

Typical cases are frequently misdiagnosed as Niemann–Pick disease. Differential diagnosis usually includes other conditions, such as Gaucher disease and other LSD, but can also consider familial hypercholesterolemia [17] and familial hemophagocytic lymphohistiocytosis [2], as well as nongenetic causes of hepatosplenomegaly, such as nonalcoholic fatty liver disease [17].

In cases of early onset (WD form), immediate intervention is crucial. Therapeutic approaches for LALD vary depending on disease severity. Despite efforts like dietary fat reduction, parenteral nutrition, anemia correction through transfusions, and drug treatments for acute conditions, the limited success of these strategies is evident. Other therapeutic possibilities include liver transplant (LT), enzyme replacement therapy (ERT; sebelipase alfa recombinant protein, Kanuma^®^, Kanuma, Japan), and hematopoietic stem cell transplant (SCT). Therapies based on mRNA and adeno-associated virus (AAV)-mediated gene therapy aim to promote functional *LIPA* expression in patients and represent promising proposals for the future, although they are still in the research and development phase [1].

Regarding the LT strategy, Ambler et al. (2012) [20] reported the success of this procedure in a 42-year-old woman with CESD, which notably improved survival. Similarly, Sreekantam et al. (2016) [21] documented a case of a pediatric patient who underwent LT and exhibited normal growth and liver function 8 years post-transplant. Ramakrishna (2022) [22] also described a successful LT in a 16-year-old boy with LALD, with positive outcomes observed at the 10-month follow-up. However, Bernstein et al. (2018) [7] reviewed the literature on 18 cases of patients with LALD who underwent LT and described multisystemic LALD progression in 11 patients and death in 6.

Despite case reports documenting that LT in patients with LALD leads to immediate normalization of the lipoprotein profile and partial improvement of the systemic phenotype due to the secretion of LAL from healthy hepatocytes and LAL uptake by enzyme-deficient peripheral cells, challenges persist regarding disease recurrence and extrahepatic manifestations. Recurrence of CESD post-LT could also be due to the migration of deficient monocyte-derived macrophages to the liver, perpetuating the disease process. Furthermore, extrahepatic manifestations such as accelerated atherosclerosis and splenomegaly may persist or progress following LT [1,22]. There remains ongoing lipid deposition in peripheral tissues, progression of vascular lipid accumulation, renal failure, and, in some cases, a paradoxical recurrence of LALD pathology in allografts, leading to lethal outcomes [1]. These data indicate that restoring liver LAL activity alone is insufficient to ameliorate systemic pathology. However, LT is the possible treatment for severe cases with decompensated cirrhosis.

Although LT remains a therapeutic option for patients with LALD, it is a costly treatment requiring a trained team and robust healthcare infrastructure to achieve optimal outcomes. Another barrier is that it is often difficult to find compatible donors, and there is the risk of rejection and other postoperative complications [23]. The long-term management of post-LT involves comprehensive monitoring for disease recurrence and cardiovascular complications. Regular echocardiography, electrocardiography, and cardiac stress testing are recommended to assess cardiovascular risk.

Hematopoietic SCT is a strategy adopted for various LSDs, such as LALD. The principle of this type of therapy is that these cells replace bone marrow tissue and can migrate to various organs, continuously producing the deficient enzyme [24]. However, allogeneic hematopoietic SCT requires conditioning chemotherapy for bone marrow ablation in addition to immunosuppression [25]. For LALD, initial hematopoietic SCT were largely unsuccessful, likely due to liver toxicity caused by chemotherapeutic agents [1]. Although some patients survived beyond their first year of life, showing relevant health improvements at the time of the latest available report [26,27,28]; in most cases, the prognosis is unfavorable [1].

Recently, ERT with sebelipase alfa (Kanuma^®^) became available for the treatment of both WD and CESD [17]. After approval by the FDA in 2015 [7], at a dose of 1 mg/kg body weight every other week [17], with intravenous administration [15], sebelipase alfa has become an important tool with which to improve the quality of life in patients with CESD and to extend the survival of patients with WD [17]. Although empirical evidence is still minimal, there are studies indicating an improvement in liver transaminase levels and serum lipids, reduction in liver volume, steatosis, and fibrosis, and a considerable increase in survival [1].

It is important to emphasize that the initiation of ERT is crucial for favorable outcomes, as evidenced by a case we reported in this study, in which the diagnosis was late (at 24 years old) and, despite the treatment with sebelipase alfa having been adopted, the patient died at 28 years old. It is possible that the evolution of this patient would have been less severe if specific treatment with ERT had begun early.

Even though hepatic transplant and ERT with sebelipase alfa can be provided, the prognosis of patients with LALD can still be very limited if the diagnosis remains late overall.

Despite being a reference for LT over the years, Brazil still lacks descriptions of patients with LALD who underwent such procedures, which remains unusual. Further research is warranted to explore novel therapeutic approaches and improve long-term outcomes in this patient population.

There are 150 pathogenic/likely pathogenic variants in *LIPA* described in the ClinVar [29] database, with at least 39 different missense variants. The splice variant c.894G>A is present in nearly half of patients with CESD (most of them from European populations) [30]; while in individuals of Iranian Jewish origin, the c.260G>T(p.G87V) variant is the most frequent [31].

## 5. Conclusions

This study highlights all the challenges in diagnosing and treating LALD, a rare condition, that must be considered for children undergoing LT, especially those without a known cause for hepatic insufficiency. For patients presenting with hepatomegaly, elevated liver function markers, and dyslipidemia, LALD should be considered as a diagnostic hypothesis. However, due to its rarity, it is not surprising that physicians may not be sufficiently familiar with the condition to provide appropriate diagnostic and therapeutic guidance. The determination of LAL enzyme activity and the identification of the *LIPA* causal variant are essential for confirming the diagnosis.

It is necessary to alert pediatricians, hepatologists, and pediatric surgeons that this very rare genetic condition is associated with a 25% recurrence risk for future offspring of the patients’ parents, which is relevant for genetic counseling.

This case report arises from a cohort study in which, out of 73 patients enlisted for liver transplant testing, only one case of LALD was identified. Due to the late diagnosis, the treatment with sebelipase alfa was initiated very late, which, unfortunately, did not prevent the patient’s death. Given the availability of specific therapy with enzyme replacement, early and precise diagnosis has become crucial to provide the recommended care effectively.

## Figures and Tables

**Figure 1 ijms-25-08648-f001:**
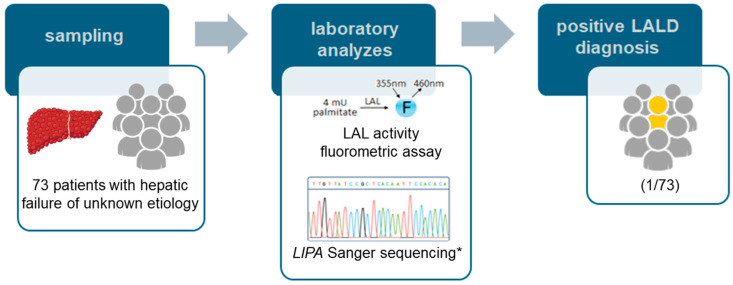
Methodological workflow for diagnosing the case described in this study. 4 mu palmitate: the substrate 4-methylumbelliferyl palmitate; LAL: lysosomal acid lipase; 355 nm and 460 nm: excitation and emission wavelengths, respectively; F: fluorescence emission; LALD: lysosomal acid lipase deficiency. * *LIPA* Sanger sequencing was only performed for patients with low LAL activity.

**Figure 2 ijms-25-08648-f002:**
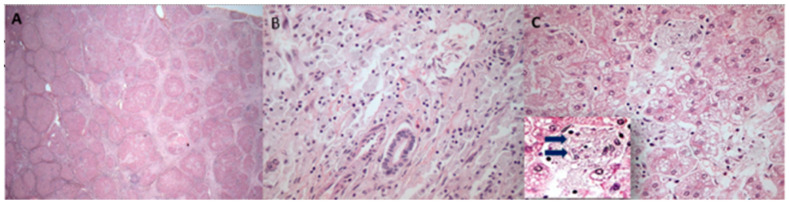
Histopathological analysis (H&E staining) of explanted liver from the patient at 18 years old: (**A**) the cirrhotic liver explant showing advanced-stage fibrosis forming micronodules (H&E, 25×); (**B**) a large number of macrophages with foamy and light-tan cytoplasm seen in portal tracts (H&E, 200×); (**C**) areas with microvesicular and mediovesicular steatosis in hepatocytes and aggregates of enlarged xanthomatous Kupffer cells (H&E, 200×), which have rare needle-shaped clefts (arrows).

**Figure 3 ijms-25-08648-f003:**
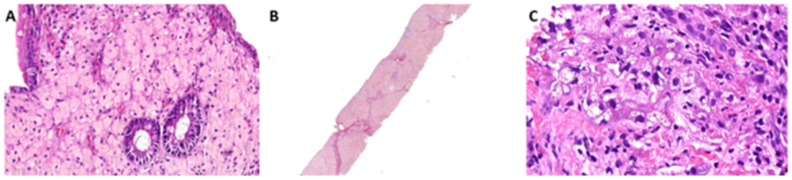
Histological analysis of new liver biopsy from the patient indicating recurrence of the storage disorder: (**A**) the duodenal mucosa sample showing diffuse infiltration by foamy macrophages (H&E, 200×); (**B**) at low magnification, the Sirius Red stain reveals architectural distortion with incomplete cirrhosis in the liver biopsy performed 7 years after the first liver transplant; (**C**) the presence of numerous foamy portal macrophages found in that biopsy, suggesting the recurrence of CESD in the graft (H&E, 400×).

## Data Availability

Prospective data have been collected from patients over the last decade. For data requests, please contact the corresponding author.

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
