# Peer review of "Evaluation of 73 Enlisted Patients for Liver Transplant with Unknown Etiology Reveals a Late-Diagnosed Case of Lysosomal Acid Lipase Deficiency"

_ijms, 2024, doi:10.3390/ijms25168648_

Round 1

Reviewer 1 Report

Comments and Suggestions for Authors

I was very pleased to read and review the manuscript written by de Medeiros Bastos and co-authors entitled „Very Late Diagnosis of Lysosomal Acid Lipase Deficiency 2 among 73 Patients Enlisted for Liver Transplant from Unknown 3 Etiology“.

From my point of view, this is a case report of a patient with early onset LALD and reoccurence of the disease in the transplanted liver, with a literature review. I have to comment that it is unclear from the title that only one patient among 73 was diagnosed with LALD. Also, I wouldn't say that the main objective of this article is the screening for LALD in the high-risk population. Therefore, the title does represent the content of the article.

Regarding the text, I have several comments and suggestions:

In the abstract – I wouldn't say that 73 patients with liver failure were possible LALD patients. They were patients tested for LALD because they met the clinical criteria for testing. LAL analysis and LIPA gene sequencing are not needed only for genetic counseling but also for establishing the right diagnosis and specific treatment. Of course, genetic counseling is also important for family planning and identifying other family members with the same disease, but more important is the right diagnosis and treatment of the index patient.  

In the introduction, Line 39 – instead of „infants“ please use the word infantile for the severe form of LALD

Cohort study, Line 73 - It is written that patients underwent extensive diagnostic evaluation including for inborn errors of metabolism. Please write which metabolic tests were done or for which diseases. This comment is because LALD is one of the metabolic diseases presenting with liver failure.

 In case report: 

For liver enzymes please add units. Also, please provide concentrations of total cholesterol, LDL, HDL, and triglycerides when mentioning hypercholesterolemia.

Line 104 – Is the meaning of „microgoticular“ microvesicular?

Line 109 – In which sample was the activity of sphingomyelinase measured? Liver, blood, fibroblasts? Please add this information. Also, add units for the activity of enzyme and acid phosphatase.

Line 125 – Instead of postmortem donor please use the word cadaveric.

Line 130 – Which genetic condition? Probably, LALD considering the finding of cholesteryl crystals.

In line 142 you mention the activity of chitotriosidase for the first time and say that levels stabilized, please provide the values with units and reference range. You are also mentioning LAL levels as persistently low. Is this lysosomal acid lipase activity? If yes, you had the diagnosis of LALD at that point already? Please explain.

Line 148 increase in transaminases didn't result in liver biopsy but probably prompted one, there are other examples in the text where chosen words are not optimal, possibly due to differences in languages. I suggest asking a native English speaker to review the text for language improvements.

Line 156 - Why did it take several years, after the finding of low LAL activity, to perform LIPA gene analysis? If the diagnosis had been made earlier and ERT treatment commenced, the outcome might have been different. Especially considering that this patient was found through the screening on LALD which is one of the objectives of this article.

Line 161 – Details (criteria and arguments) about the classification of the found LIPA variant are not necessary within the main text. I would rather add it as a supplementary material together with Figure 4.

Line 196 – Please provide information if the dose of Sebelipase was standard one and if the patient received it regularly.

Line 203 – Which acute posttransplant complications were the cause of death? On which posttransplant day patient deceased?

Line 248- Not only HSCT is effective for treating Wolman disease but also early ERT, at least initially. Some authors report the best outcome after multimodal treatment ERT + HSCT.

Line 256 – Niemann Pick types A and B are part of the spectrum of acid sphingomyelinase deficiency and not two different diseases, so no need to mention them separately in the differential diagnosis.

The discussion is mainly a literature review, which is acceptable, but as I suggested already, the title should be changed.

Comments on the Quality of English Language

I am not a native English speaker and I have issues with my English as well, so I am not the best person to comment on this, but I think that minor editing of English language would improve the manuscript. 

Author Response

Please find attached the rebuttal letter with responses to both reviewers.

Let me know if you'd like any further adjustments!

Reviewer 2 Report

Comments and Suggestions for Authors

Reviewer Comments

Very Late Diagnosis of Lysosomal Acid Lipase Deficiency among 73 Patients Enlisted for Liver Transplant from Unknown Etiology

Overall, this paper presents useful information about a disorder associated with a missense variant in LIPA. There are, however, a number of weaknesses that should be addressed: (1) concurrent Niemann-Pick disease is not ruled out; (2) the unaffected parents and sibling of the proband should be genotyped for the LIPA variant found in the proband; and (3) the disease phenotype of the proband should be compared to that of the subject with the same homozygous variant reported by Benevides et al.

Specific suggestions for improving the manuscript are as follows:

Abstract

Line 24: Change “submitted to” to “that underwent”.

Line 29: Change “been submitted to” to “undergone”.

Line 32: Change "diagnose” to “diagnosis”.

Introduction

Lines 56-58: What was the basis for concluding that the Niemann-Pick type B diagnosis was erroneous?

Case report

Lines 101-102: What is meant by blood cells being normal? Normal in what ways?

Lines 152-155 and Figure 3: Why was the liver pathology interpreted as being due to recurrence of the storage disorder and not to rejection of the transplanted liver? The donor liver presumably had normal LAL activity.

Figures 3 and 4: Need arrows pointing to pathological features described in the figure captions.

Lines 191-193: The authors measured enzyme activity using an artificial substrate and determined that the enzyme activity in the proband was zero. However, they need to consider the possibility that residual enzyme activity with natural substrates remained. It would help the case that the LIPA variant was responsible for the disorder if the genotypes of the presumably unaffected parents and sibling were provided. The study does not rule out the possibility that the proband had concurrent Niemann-Pick disease.

Line 194: Indicated what Sebelipase alfa is.

Lines 197-199: See comment for lines 152-155.

Discussion

Lines 233-234: As stated above, the study does not rule out the possibility that the proband had concurrent Niemann-Pick disease.

Line 239: The authors should compare the disease phenotype of their proband with that of the subject described by Benevides et al.

Lines 244-249: The authors report that the LAL activity in the proband was zero, but the subject did not exhibit the early-onset, more severe disease phenotype. This needs to be discussed.

Lines 293-296: Grammar needs correction in these sentences.

Line 298: Need to define the abbreviation SCT

Comments on the Quality of English Language

Minor edits have been suggested in my review.

Author Response

(The authors gave the same response as above.)
